# Phosphorylated tau181 in plasma as a potential biomarker for Alzheimer's disease in adults with Down syndrome

Alberto Lleó [1,2,13✉], Henrik Zetterberg [3,4,5,6,13], Jordi Pegueroles[1,2], Thomas K. Karikari [3], María Carmona-Iragui[1,2,7], Nicholas J. Ashton[3,8,9,10], Victor Montal[1,2], Isabel Barroeta[1,2], Juan Lantero-Rodríguez [3], Laura Videla[1,2,7], Miren Altuna [1,2], Bessy Benejam[1,2,7], Susana Fernandez[7], Silvia Valldeneu[1,2], Diana Garzón[1,2], Alexandre Bejanin [1,2], Maria Florencia Iulita[1,2], Valle Camacho [1], Santiago Medrano-Martorell [11], Olivia Belbin [1,2], Jordi Clarimon[1,2], Sylvain Lehmann[12], Daniel Alcolea [1,2], Rafael Blesa[1,2], Kaj Blennow [3,4,14] & Juan Fortea[1,2,7,14✉]

Plasma tau phosphorylated at threonine 181 (p-tau181) predicts Alzheimer's disease (AD) pathology with high accuracy in the general population. In this study, we investigated plasma p-tau181 as a biomarker of AD in individuals with Down syndrome (DS). We included 366 adults with DS (240 asymptomatic, 43 prodromal AD, 83 AD dementia) and 44 euploid cognitively normal controls. We measured plasma p-tau181 with a Single molecule array (Simoa) assay. We examined the diagnostic performance of p-tau181 for the detection of AD and the relationship with other fluid and imaging biomarkers. Plasma p-tau181 concentration showed an area under the curve of 0.80 [95% CI 0.73–0.87] and 0.92 [95% CI 0.89–0.95] for the discrimination between asymptomatic individuals versus those in the prodromal and dementia groups, respectively. Plasma p-tau181 correlated with atrophy and hypometabolism in temporoparietal regions. Our findings indicate that plasma p-tau181 concentration can be useful to detect AD in DS.

[1] Memory Unit, Department of Neurology, Hospital de la Santa Creu i Sant Pau, Biomedical Research Institute Sant Pau, Universitat Autònoma de Barcelona, Barcelona, Spain. [2] Centro de Investigación Biomédica en Red Enfermedades Neurodegenerativas (CIBERNED), Madrid, Spain. [3] Department of Psychiatry and Neurochemistry, Institute of Neuroscience and Physiology, the Sahlgrenska Academy at the University of Gothenburg, Möndal, Sweden. [4] Clinical Neurochemistry Laboratory, Sahlgrenska University Hospital, Mölndal, Sweden. [5] UK Dementia Research Institute at UCL, London, UK. [6] Department of Neurodegenerative Disease, UCL Institute of Neurology, London, UK. [7] Barcelona Down Medical Center, Fundació Catalana Síndrome de Down, Barcelona, Spain. [8] Wallenberg Centre for Molecular and Translational Medicine, University of Gothenburg, Gothenburg, Sweden. [9] Department of Old Age Psychiatry, Institute of Psychiatry, Psychology and Neuroscience, King's College London, London, UK. [10] NIHR Biomedical Research Centre for Mental Health and Biomedical Research Unit for Dementia at South London and Maudsley NHS Foundation, London, UK. [11] Hospital del Mar-Universitat Autònoma Barcelona (UAB), Barcelona, Spain. [12] The Institute for Neurosciences of Montpellier, Université de Montpellier, Centre Hospitalier Universitaire de Montpellier, INSERM, Montpellier, France. [13] These authors contributed equally: Alberto Lleó, Henrik Zetterberg. [14] These authors jointly supervised this work: Kaj Blennow, Juan Fortea. ✉email: alleo@santpau.cat; jfortea@santpau.cat

Down syndrome (DS) is the most frequent form of developmental intellectual disability of genetic origin, affecting 5.8 million people worldwide[1]. Due to improved medical care, life expectancy has dramatically increased, currently exceeding 60 years of age[2]. However, age-related comorbidities have also emerged. In particular, the lifetime risk of Alzheimer's disease (AD) in people with DS is over 90%[3,4]. As a result, AD has become the main cause of death in this population[5]. In spite of the strong association between DS and AD, this comorbidity is still underrecognized and underdiagnosed[1], which prevents or delays access to appropriate medical care and clinical trials.

We and others have described the usefulness of cerebrospinal fluid (CSF) and amyloid positron emission tomography (PET) biomarkers to detect the core neuropathological hallmarks of AD in this population[5–8]. All these studies support the notion that AD in DS recapitulates the pattern observed in both sporadic and autosomal-dominant AD. However, despite the high accuracy of core CSF and imaging biomarkers to detect AD, the costs and low accessibility of these tests may limit the broad application in clinical routine, restricting their use to specialized centres. In recent years, blood biomarkers have emerged as an easy and cost-effective alternative for the screening of AD[9]. In particular, plasma NfL and the ratio $A\beta_{1-42/1-40}$ have been described as accurate peripheral markers to detect neurodegeneration and brain amyloidosis, respectively[10–12]. However, changes in plasma NfL are not specific to AD[10], and changes in plasma $A\beta$ levels (and the ratio $A\beta_{1-42/1-40}$) are small compared to the change in CSF[13], which confers high demands on between-assay precision for clinical routine. A recent advance has been the description of plasma p-tau assays, mainly p-tau181 and p-tau217, that can differentiate AD from other neurodegenerative diseases with high accuracy[14–19]. Currently, only one small study[20] has investigated plasma p-tau181 levels in 20 adults with DS. However, the method lacked analytical sensitivity to measure plasma p-tau in around half of DS cases, and that study did not assess the AD clinical status or the diagnostic performance of this biomarker.

In this cross-sectional study, we assessed the accuracy of p-tau181 in plasma to detect AD in a large cohort of adults with DS and describe the association with other biochemical and neuroimaging AD biomarkers.

## Results

**Participants**. We included 366 adults with DS and 44 euploid controls. Table 1 summarizes the demographics, cognitive and biomarker data. 170 (46.4%) of the participants with DS and 23 (52.3%) of the controls were female. Among the participants with DS, 21.9% had mild intellectual disability, 51.1% moderate and 27.0% severe or profound. Of the participants with DS, 240 (65.6%) were asymptomatic, 43 (11.7%) had prodromal AD and 83 (22.7%) had AD dementia at the time of blood draw. The median age of diagnosis was 50.3 [48.0–54.1] for prodromal AD and 53.2 [49.2–57.1] for AD dementia. As previously reported[4,7], participants with DS and prodromal AD and AD dementia showed a decrease in the CSF ratio of $A\beta_{1-42/1-40}$ and an increase in CSF concentration of total tau, p-tau181 and NfL, as well as an increase in plasma concentration of NfL compared with asymptomatic subjects and euploid controls (Table 1).

**Group comparisons and diagnostic accuracy of p-tau181.** While plasma p-tau181 concentration was relatively stable in the euploid controls across age, they increased in the early thirties in DS (Fig. 1), and by the age of 40.5 years, the confidence intervals of adults with DS and euploid controls did not overlap (10 years before expected prodromal AD diagnosis).

There were no differences in plasma p-tau181 concentration between asymptomatic DS participants and controls (11.36 pg/mL vs 9.31 pg/mL; $p = 0.139$; Fig. 2A, Table 1). Median plasma p-tau181 concentration in participants with DS and prodromal AD and AD dementia were increased approximately two-fold compared to those of asymptomatic participants with considerable overlap (Fig. 2A, Table 1; $p < 0.0001$). Plasma p-tau181 concentration in participants with DS and AD dementia were higher compared to those of participants with prodromal AD (32.58 pg/mL vs 21.72 pg/mL; $p = 0.024$, Fig. 2A, Table 1). Plasma p-tau181 concentration was higher in participants with moderate or severe/profound intellectual disability than in those with mild intellectual disability (Supplementary Fig. 1). We next analysed the accuracy of plasma p-tau181 concentration for the diagnosis of AD in DS (Fig. 2B). The AUC was 0.80 [95% CI 0.73–0.87] for the comparison between asymptomatic individuals versus those with prodromal AD and 0.92 [95% CI 0.89–0.95] for the comparison between asymptomatic individuals versus those with AD dementia. The AUC was 0.88 [95% CI 0.84–0.91] for the comparison between asymptomatic individuals versus those with symptomatic AD (prodromal and dementia groups combined). The diagnostic performance of p-tau181 increased in combination with age and *APOEε4* status (Supplementary Fig. 2). For comparison purposes, diagnostic performance of CSF biomarkers is shown in Supplementary Fig. 3.

**Comparisons of plasma p-tau181 and plasma NfL**. We next compared the diagnostic accuracy of plasma p-tau181 levels with the levels in plasma of NfL in the subset of participants with both measures (Fig. 2C, Supplementary Table 1, $n = 289$)[7]. The AUC for the comparison between asymptomatic individuals versus those with prodromal AD was 0.88 [95% CI 0.82–0.93] for p-tau181 and 0.86 [95% CI 0.81–0.92] for NfL. The AUC for the comparison between asymptomatic individuals versus those with AD dementia was 0.94 (95% CI 0.91–0.97) for p-tau181 and 0.96 [95% CI 0.93–0.98] for NfL (Fig. 2C and D). The AUC was 0.92 [95% CI 0.89–0.95] for plasma p-tau181 and 0.93 [95% CI 0.90–0.96] for plasma NfL for the comparison between asymptomatic individuals versus those with symptomatic AD (prodromal and dementia). The differences in diagnostic accuracy between plasma p-tau181 and plasma NfL were not statistically significant.

**Associations with other fluid biomarkers**. We also analysed the correlation between log-transformed plasma p-tau181 levels and other core AD fluid biomarkers (Supplementary Fig. 2). In subjects with DS plasma p-tau181 concentration correlated with plasma NfL concentration (rho = 0.70; $p < 0.0001$). In paired plasma-CSF samples there was a correlation between plasma p-tau181 concentration and the CSF ratio $A\beta_{1-42/1-40}$ (rho = −0.52; $p < 0.0001$), CSF concentration of total tau (rho = 0.63; $p < 0.0001$), CSF concentration of p-tau181 (rho = 0.68; $p < 0.0001$) and CSF concentration of NfL (rho = 0.58; $p < 0.0001$, Supplementary Fig. 4).

**Associations with imaging biomarkers**. Next, we analysed the correlation between plasma p-tau181 and imaging biomarkers in participants with DS. Plasma p-tau181 concentration correlated with atrophy measured by MRI in characteristic AD regions including the temporal regions angular and supramarginal gyri and precuneus of both hemispheres ($n = 121$, Fig. 3). These associations also extended into the lateral frontal and orbitofrontal and some occipital structures. The stratified analyses by clinical group showed that these results were primarily driven by patients with symptomatic AD. Similarly, plasma p-tau181

**Table 1 Demographics, cognitive and biomarker data of participants with Down syndrome and controls.**

|  | Control | aDS | pDS | dDS |
|---|---|---|---|---|
| n | 44 | 240 | 43 | 83 |
| Age (years) (median [IQR]) | 55.75 [47.50, 62.02]* | 37.83 [29.90, 45.58]*+& | 50.27 [48.02, 54.10]& | 53.21 [49.19, 57.14]+ |
| Gender = Male (%) | 21 (47.7) | 132 (55.0) | 22 (51.2) | 42 (50.6) |
| APOEε4 positivity (%) | 10 (23.3) | 47 (19.7) | 10 (23.3) | 19 (23.5) |
| MMSE score (median [IQR]) | 30.00 [29.00, 30.00] |  |  |  |
| CAMCOG score (median [IQR]) |  | 77.00 [62.00, 86.00]+& | 61.00 [46.50, 73.50]&^ | 41.00 [30.00, 55.00]+^ |
| Degree of disability (%) |  |  |  |  |
| Mild |  | 68 (28.3) | 7 (16.3) | 5 (6.0) |
| Moderate |  | 120 (50.0) | 18 (41.9) | 49 (59.0) |
| Severe/Profound |  | 52 (21.7) | 18 (41.9) | 29 (34.9) |
| plasma p-tau181 (pg/ml) (median [IQR]) | 9.31 [7.46, 13.82]$# | 11.36 [8.10, 16.01]+& | 21.72 [16.55, 37.02]&^ | 32.58 [23.67, 44.57]+$^ |
| plasma NfL (pg/ml) (median [IQR]) | 3.38 [2.89, 4.16]$#* | 5.93 [4.43, 10.25]+&* | 13.61 [11.50, 18.26]&^ | 23.86 [17.33, 33.65]+$^ |
| CSF Aβ42/Aβ40 (median [IQR]) | 0.11 [0.10, 0.11]$#* | 0.08 [0.06, 0.09]+&* | 0.04 [0.04, 0.05]& | 0.05 [0.04, 0.05]+$ |
| CSF t-tau (pg/ml) (median [IQR]) | 239.00 [181.75, 295.50]$# | 309.00 [163.00, 446.00]+& | 711.50 [467.25, 1112.50]& | 963.00 [653.00, 1222.00]+$ |
| CSF p-tau (pg/ml) (median [IQR]) | 32.15 [24.55, 41.52]$# | 29.80 [17.30, 51.20]+& | 111.90 [69.30, 201.07]& | 152.80 [93.40, 192.80]+$ |
| CSF NfL (pg/ml) (median [IQR]) | 362.00 [260.10, 497.56]$# | 365.00 [233.30, 507.60]+& | 705.05 [645.00, 1027.05]&^ | 1201.00 [893.58, 1625.00]+$^ |
| SUVR 18F-FBP (median [IQR]) |  | 1.15 [1.09, 1.20]+& | 1.35 [1.17, 1.43]& | 1.33 [1.30, 1.34]+ |
| SUVR 18F-FDG (median [IQR]) |  | 1.35 [1.26, 1.41] +& | 1.11 [1.07, 1.30]&^ | 0.82 [0.74, 0.93]+^ |

*N sample, IQR interquartile range, ID intellectual disability, CSF cerebrospinal fluid, Aβ Amyloid-β, NfL neurofilament light protein, FDG 18-fluorodeoxyglucose, SUVR Standardized Uptake Value Ratio. aDS asymptomatic Down syndrome, pDS prodromal Alzheimer's disease Down syndrome, dDS Alzheimer's disease dementia Down syndrome, Controls euploid healthy controls. Symbols designate significant differences between groups: control-aDS (*), control-pDS (#), control-dDS ($), aDS-pDS (&), aDS-dDS (+) and pDS-dDS (^).*

concentration correlated with lower brain metabolism in temporoparietal regions measured by FDG-PET ($n = 68$), and this association was mainly driven by patients with symptomatic AD (Fig. 3B). The areas with hypometabolism overlapped with the areas with atrophy.

Finally, we compared concentration of plasma p-tau181 in participants with amyloid PET available. Of the 45 DS cases with amyloid PET, 10 were amyloid negative (SUVr < 1.11). The mean plasma p-tau181 concentration was higher in participants with DS that had a positive amyloid PET scan compared with those with a negative amyloid status (Fig. 4A). The AUC for the comparison between both groups was 0.77 (95% CI 0.61–0.93; Fig. 4B).

## Discussion

This study describes that plasma p-tau181 can be reliably used for the screening of AD in a well-characterised cohort of participants with DS. Plasma p-tau181 concentration start to increase in the early thirties and correlate with core fluid biomarkers of AD as well as with cortical thinning and brain hypometabolism in AD-related brain regions.

Recent biomarker studies[14–17,19,20] have shown that blood p-tau181 can identify central tau pathology with high accuracy in sporadic AD. Concentration of p-tau181 in blood is increased in individuals with AD with respect to cognitively healthy individuals, they predict amyloid β PET positivity and are not elevated in non-AD conditions[14–17,19]. In addition, concentration of p-tau181 in blood correlate with CSF p-tau181, tau pathology measured by PET and Braak stages post-mortem[15]. There is only one small study with 20 adults with DS that has investigated plasma p-tau181 concentration[20]. However, this study did not assess the diagnostic performance, did not evaluate the clinical dementia status of the participants and the method had insufficient analytical sensitivity.

Our data in individuals with DS are in good agreement with recent studies on plasma p-tau-181 in sporadic and autosomal-dominant AD[14–17,21]. In DS, changes in p-tau181 concentration in blood are already detectable in the fourth decade with a very similar temporality and magnitude of change to plasma NfL concentration and CSF p-tau181, total tau and NfL levels[4]. The pattern of change of plasma p-tau181 in DS is also very similar to that described in autosomal-dominant AD[21]. Similarly to CSF total tau and CSF NfL concentrations[4], plasma p-tau181 concentration was similar in asymptomatic adults with DS and controls, but showed a two-fold increase in median plasma p-tau concentration in patients with prodromal AD and AD dementia compared with asymptomatic individuals. This result is in disagreement with the only other study to have investigated plasma p-tau181 concentration in DS[20], which found higher levels in the 20 subjects with DS with respect to the 22 controls. The group differences might have been driven by undiagnosed symptomatic patients as the study did not asses dementia status and all adults with DS and high plasma p-tau181 concentrations were older than 40. In contrast, we show that young adults with DS have normal concentration of plasma p-tau181.

A relevant finding of this study is that we show a strong correlation of plasma p-tau181 and plasma NfL in adults with DS. We have shown in a previous study that plasma NfL has an AUC of 0.95 (95% CI 0.92–0.98) for the differentiation of asymptomatic vs AD dementia in Down syndrome[7]. In the present study, there were no statistical differences in accuracy between both markers. Despite the lack of disease specificity of NfL, these results can be explained by the fact that DS is a genetically determined form of AD[4,22] and that individuals with DS have a 90% lifetime risk of developing AD. In other words, a non-specific neurodegeneration marker (NfL) mirrors the diagnostic performance of a specific AD marker (p-tau181) because the other conditions that elevate NfL levels in adults with DS are exceedingly rare. In contrast, the general population is at risk of neurodegenerative dementias other than AD, which explains the different performance. Nonetheless, both in clinical practice and trials, these two biomarkers could provide complementary

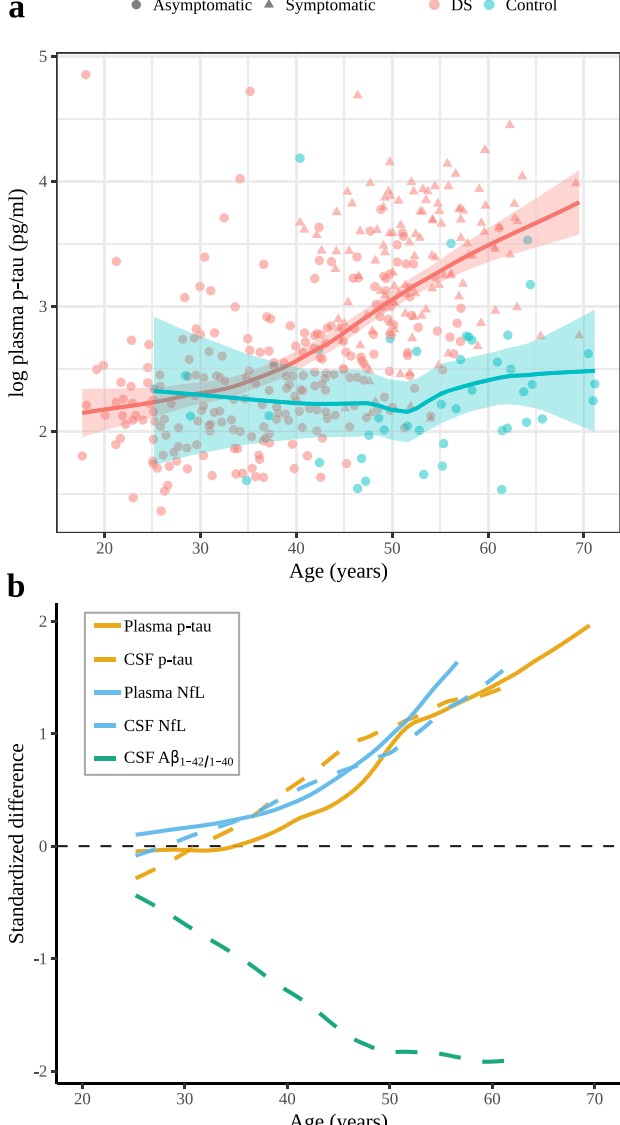

**Fig. 1 Changes with age in plasma p-tau181 concentrations in Down syndrome and euploid controls. a** Age-related changes in p-tau181 levels in individuals with Down syndrome (asymptomatic, prodromal Alzheimer's disease and Alzheimer's disease dementia, all in red) and in euploid controls (blue). The central lines indicate the fitted linear model for each group and the shadowed ribbons show the 95% confidence level intervals. **b** Integrated model of the natural history of Alzheimer' disease in Down syndrome. Comparison of the evolution of the standardized differences between participants with Down syndrome and controls fitted with a locally estimated scatterplot smoothing curve. Plasma p-tau levels are represented in a solid red line and are compared with CSF p-tau levels and both plasma and CSF NfL levels (modified from ref. 4). Positive standardized differences represent higher biomarker values in participants with Down syndrome compared to euploid controls and negative values represent lower biomarker values. Standardized differences were computed by the difference between the DS and the controls divided by the standard deviation of both groups.

information. Plasma p-tau181 could be prioritized as a first diagnostic screening, and a measure of target engagement in trials with anti-tau compounds. In turn, NfL could be used for monitoring disease progression and neurodegeneration during follow-up. In this sense, it is worth mentioning that longitudinally only

CSF NfL, but not CSF p-tau181 concentration, increase in autosomal-dominant and sporadic AD[23,24]. Future studies should assess whether the longitudinal trajectories of plasma NfL and plasma p-tau181 differ in DS (and in the general population).

One relevant difference between our study and previous studies in sporadic AD is that concentration of p-tau181 in plasma in DS predicted amyloid β PET positivity (AUC 0.77) with less accuracy than in sporadic AD (AUC 0.80–91)[14–16]. This is not entirely unexpected given the context of application in a population genetically determined to develop AD. Indeed, in the elderly euploid population only a proportion of subjects have AD pathological changes. In DS and autosomal-dominant AD, all subjects eventually develop AD pathological changes with time[4,25]. In these populations, AD biomarkers show some degree of co-linearity in the AD continuum, more so than in healthy controls[26], although with different temporality[4,25]. Therefore, the greater variability of AD pathological changes in the general population facilitates the accuracy to detect amyloid positivity. Furthermore, despite the co-linearity, the different biomarkers have different temporalities and sensitivities in the DS population with respect to other forms of AD. Thus, amyloid-β PET has indeed limited sensitivity to detect the earliest stages of amyloid deposition in DS[27], and changes in amyloid burden with PET are detected ten years later than with CSF $A\beta_{1-42/1-40}$ levels[4]. Of note, a similar offset has been reported in autosomal-dominant AD[25]. Thus, the context of application in a population with genetically determined AD and this sequence of changes, with early amyloid deposition in most individuals, likely explain the limited performance of plasma p-tau181 in predicting amyloid-β PET positivity in DS with respect to the general population.

Finally, we found that plasma p-tau181 concentration correlated with cortical thickness and brain metabolism, mainly in subjects with DS with symptomatic AD. These findings suggest that plasma p-tau181 is associated with AD-related neurodegeneration, as it has been suggested in sporadic AD[16]. If this is the case, then plasma p-tau181 could serve as a predictor of disease progression[14]. Future studies should assess if plasma p-tau181 concentration can be used to identify asymptomatic individuals with DS most likely to progress to AD dementia. It will be also important to investigate the longitudinal change of plasma p-tau and its association with the rate of decline, as well as the correlation with tau PET imaging and with post-mortem neuropathological measures of tau pathology, both intraneuronal neurofibrillary tangles and tau-positive neurites surrounding amyloid plaques.

Our results have several important implications. First, they support the use of plasma p-tau181 as an easily accessible biomarker of tau pathology in DS. Second, these results show that the changes in this biomarker are similar to those described in sporadic and autosomal-dominant AD. Finally, our findings suggest that plasma p-tau181 may be a useful biomarker in clinical trials in DS, either for screening or inclusion purposes, or to monitor drug effects on tau pathology in clinical trials in DS. People with DS are a suitable population to conduct AD clinical trials, and the prevalence is higher than other genetic-determined forms of AD. Unfortunately, despite the remarkably high risk to develop symptomatic AD in this population, very few trials have been performed in individuals with DS. We previously showed that a significant proportion of adults with DS are capable and willing to perform all the multimodal studies required in a trial[4]. The implementation of easily accessible biomarkers, such as plasma p-tau181 and NfL, may accelerate the testing of therapies for AD in people with DS.

The main strengths of this study are the large sample size with a wide range of age and the comprehensive and multimodal nature of the evaluation that includes clinical, biochemical and

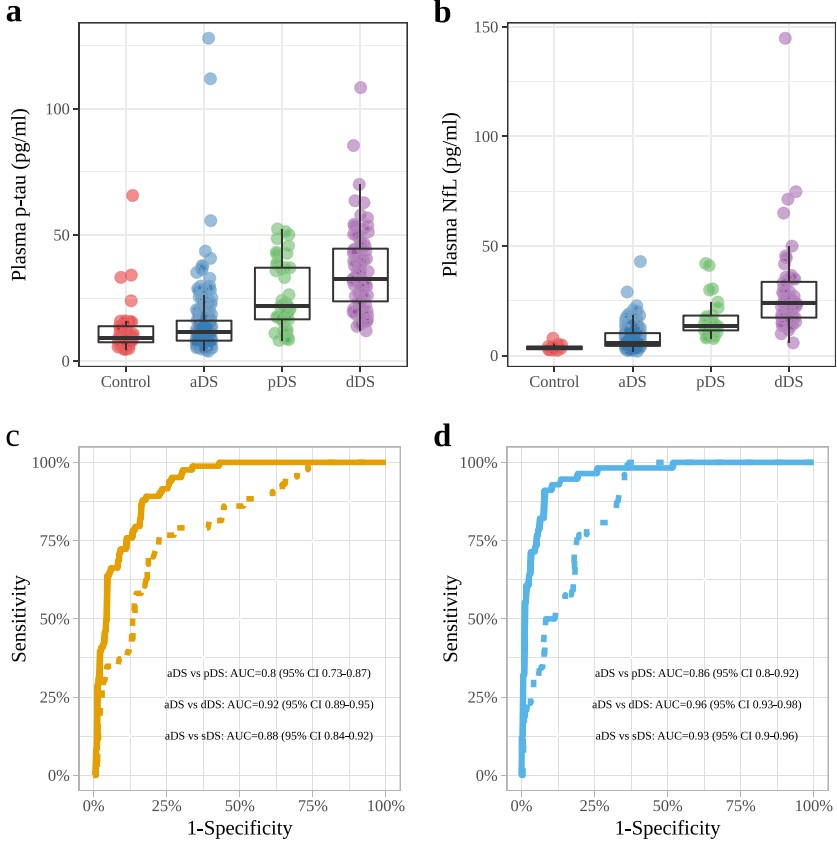

**Fig. 2 Plasma p-tau181 and NfL concentrations in Down syndrome clinical groups and controls. a**, **b** Box and whisker plots of the median concentrations of plasma p-tau181 and plasma NfL. Plasma p-tau181 concentrations **a** for aDS ($n = 240$), pDS ($n = 43$), dDS ($n = 83$) and euploid controls ($n = 44$) and plasma NfL concentrations **b** for aDS ($n = 193$), pDS ($n = 26$), dDS ($n = 56$) and euploid controls ($n = 14$). The central black lines indicate the median values. The boxes above and below these lines show the upper and lower quartiles, respectively, and the whiskers illustrate upper and lower 1.5× IQR limits. **c**, **d** ROC curves for plasma p-tau181 (**c**) and NfL (**d**) comparing the asymptomatic group with the Alzheimer's disease dementia (full curve) and with the prodromal Alzheimer's disease group (dotted curve). ROC receiver operating characteristic, NfL neurofilament light protein, p-tau tau phosphorylated at threonine 181.

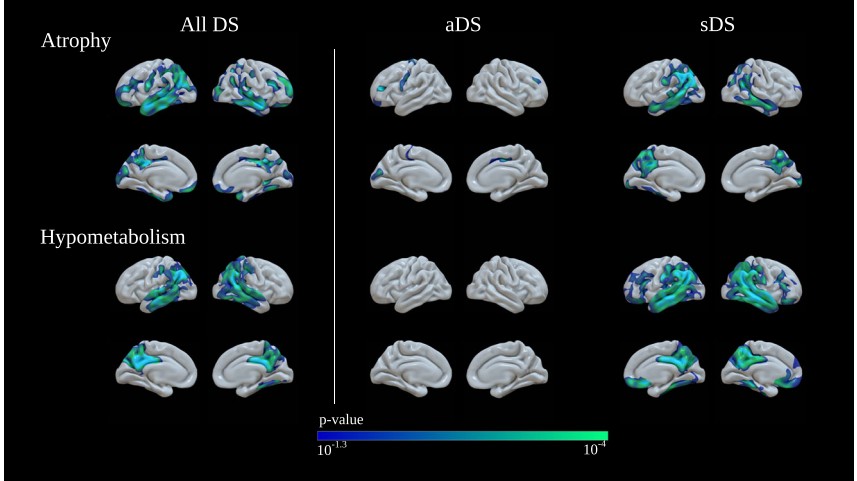

**Fig. 3 Association of plasma p-tau181 levels with imaging biomarkers in Down syndrome. a** Association of p-tau181 levels with cortical thickness measured by MRI in Down syndrome subjects. Levels of plasma p-tau181 correlated with atrophy in Alzheimer's disease typical regions. **b** Association of plasma p-tau181 levels with brain metabolism measured by [18]FDG-PET in Down syndrome subjects. Levels of plasma p-tau181 correlated with lower brain metabolism, also driven by patients with symptomatic Alzheimer's disease. aDS asymptomatic Down syndrome, sDS Down syndrome with symptomatic Alzheimer's disease.

**a**

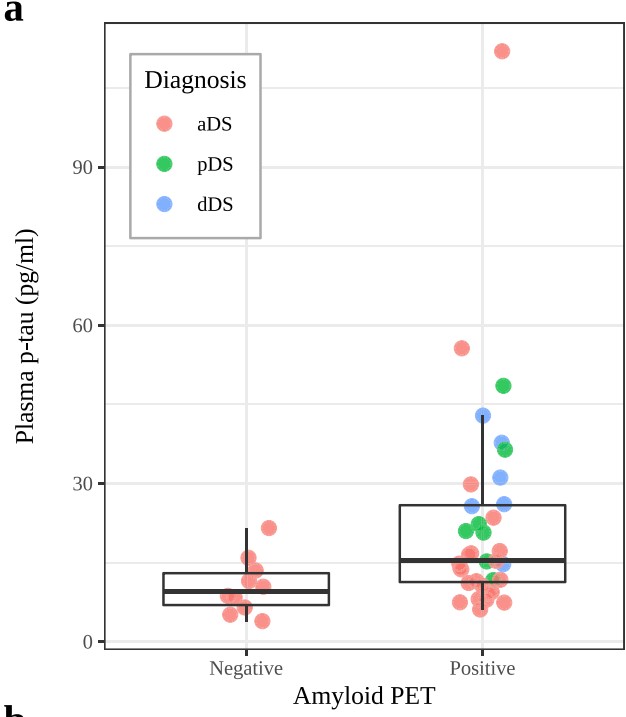

**b**

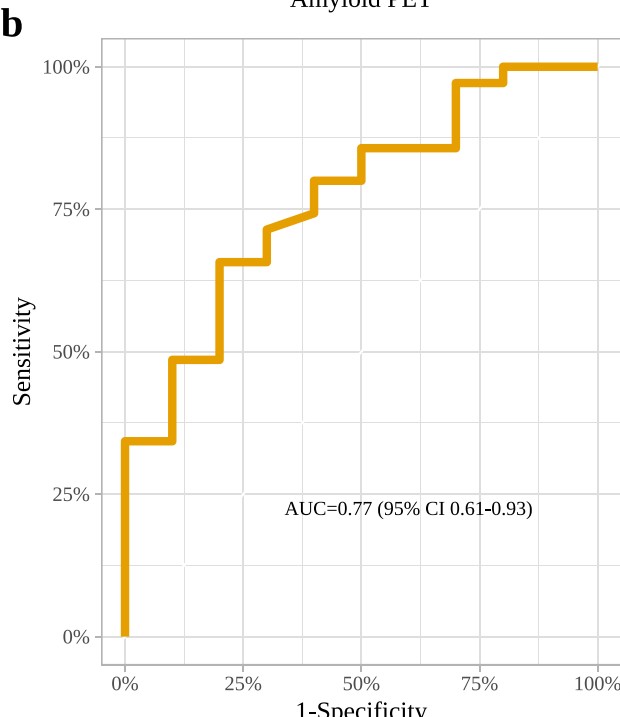

**Fig. 4 Plasma p-tau181 levels to predict Amyloid-β PET positivity in Down syndrome. a** Plasma p-tau181 levels for Down syndrome subjects stratified by amyloid-β PET status (n PET negative = 0; n PET positive = 35). The central black lines indicate the median values. The boxes above and below these lines show the upper and lower quartiles, respectively, and the whiskers illustrate upper and lower 1.5× IQR limits. **b** ROC curves for plasma p-tau181 comparing the individuals with Down syndrome with positive and negative amyloid-β PET. ROC receiver operating characteristic, p-tau tau phosphorylated at threonine 181.

imaging markers. The main limitations are the cross-sectional design, the lack of tau PET imaging or pathological confirmation and that the sample sizes differ between biomarker modalities.

In conclusion, our study shows that p-tau181 in plasma can be useful for the detection of AD in DS. This biomarker correlates with the characteristic hallmarks of AD observed in other biomarker modalities. Determinations of plasma p-tau181 may be useful as a first screening tool to detect AD in DS, particularly in those with severe intellectual disability where a precise clinical diagnosis is more difficult, as well as in and clinical trials.

## Methods

**Study design and participants.** We performed a single-centre cross-sectional study of adults with Down syndrome and euploid controls in Barcelona (Spain). Adults with DS were recruited from a population-based health plan designed to screen for AD dementia, which includes yearly neurological and neuropsychological assessments. Those individuals interested in research studies are included in the Down Alzheimer Barcelona Neuroimaging Initiative (DABNI) cohort[4,7]. We recruited the euploid cognitively normal controls from the Sant Pau Initiative on Neurodegeneration (SPIN) cohort[28]. The period of recruitment was February 2013 to December 2019.

The study was approved by the Ethical Review Board of the Sant Pau Research Institute, following the standards for medical research in humans recommended by the Declaration of Helsinki. All participants or their legally authorized representatives gave written informed consent before enrolment. We included all adults with DS that had plasma available.

**Procedures.** For the purpose of dementia diagnosis, we administered a semi-structured adapted health questionnaire to the caregivers, the Cambridge Examination for Mental Disorders of Older People with Down Syndrome and others with intellectual disabilities (CAMDEX-DS) developed in Cambridge (UK), and also adapted to the Spanish population. The CAMDEX includes a comprehensive battery covering seven different cognitive domains, the Cambridge Cognitive Examination for Older Adults with Down Syndrome (CAMCOG-DS). Using these tools, we classified participants with Down syndrome into asymptomatic (no clinical or neuropsychological suspicion of AD), prodromal AD (suspicion of AD, but symptoms do not fulfill criteria for dementia) or AD dementia (subjects with DS with full-blown dementia) in a consensus meeting between the neurologist/psychiatrist and the neuropsychologists who assessed them blind to the biomarker data[4,7]. We stratified the level of intellectual disability according to the Diagnostic and Statistical Manual of Mental Disorders, Fifth Edition as mild, moderate, severe or profound (which were grouped together). Classification was based on the individuals' best-ever level of functioning. The information was obtained through family interview and review of medical or educational records for past assessment results.

Euploid controls underwent a structured neurological assessment and a comprehensive neuropsychological battery. Inclusion criteria were normal neuropsychological results for their age and education, a clinical dementia rating scale score of 0 and normal levels of core AD biomarkers in CSF[28].

Genetic screening of trisomy 21 was assessed in 308 adults with DS (75.3%), and APOEε4 carrier status was obtained following previously published protocols[4,7].

A subset of participants had 3T-MRI (n = 121) and/or 18F-fluorodeoxyglucose PET (FDG-PET, n = 68) and/or 18F-Florbetapir PET acquisitions (n = 45) available for analysis[4,7]. Structural T1 MRI was processed using the cortical reconstruction pipeline of Freesurfer v6 (https://surfer.nmr.mgh.harvard.edu/). The estimated cortical thickness (CTh) individual surfaces were inspected in order to detect segmentation errors, and a smoothing kernel of 15 mm was applied to all the images. FDG-PET images were co-registered to the individual MRI and intensity scaled by the uptake of the pons-vermis region and the resulting images were projected to the middle point of the cortical ribbon. Finally, these images were visually inspected for possible errors and a smoothing kernel of 10 mm was then applied[4]. For the 18F-Florbetapir scans, whole cerebellum was used as reference region[29]. The cut-off point for amyloid positivity was 1.11[29].

CSF and blood samples were acquired concurrently on the same day following established procedures[14]. Briefly, CSF samples were collected in 10 mL polypropylene tubes (Sarstedt, Ref#62.610.018) and transferred to the Sant Pau Memory Unit's laboratory where they were processed and aliquoted within the first 2 h after the lumbar puncture and stored at −80 °C until analysis. Plasma samples were collected in 10 ml ETDA tubes, and centrifuged at 4 °C for 10 min, and aliquoted within the first 2 h in 100 µl tubes. The median storage time was 3.5 years (IQR 1.7–4.4) and all samples had a maximum of two freeze/thaw cycles before the analysis. Plasma levels of p-tau181 were measured with a validated Single molecule array (Simoa) immunoassay, as described previously in detail[13], by technicians blinded to the biomarker and clinical data at the Sahlgrenska Academy at the University of Gothenburg, Möndal, Sweden. Plasma levels of NfL were measured using the commercially available NF-light kit (Quanterix, Billerica, MA) by Simoa at Centre Hospitalier Universitaire Montpellier (n = 289, Montpellier, France). CSF was available in a subset of participants (n = 129). CSF levels of Aβ1-42, Aβ1-40,

p-tau181 and total tau were quantified in all samples using a commercially available immunoassay in the fully automated platform LUMIPULSE (Fujirebio-Europe, Ghent, Belgium)[29]. CSF NfL levels were measured with a commercially available ELISA (UmanDiagnostics, Umeå, Sweden) following the manufacturer's recommendations. All CSF samples were analysed at Hospital Sant Pau. Methods and results from plasma NfL and CSF biomarkers other than plasma p-tau181 in this cohort have already been published[4,7]. All measurements were obtained from distinct samples.

**Statistical analysis**. All the statistical analyses were performed using R statistical software version 3.6.3. Baseline characteristics were reported using standard descriptive statistics. Continuous variables were indicated as median [IQR] and categorical data were summarised as absolute frequencies and percentages. Differences in baseline characteristics between the diagnostic groups were assessed using Kruskal–Wallis and Dunn's tests with Holm multiple comparisons correction. To determine the temporality of plasma p-tau181 changes in DS we fitted a 1st degree locally estimated scatterplot smoothing curve in controls and in adults with DS independently[4]. We defined biomarker change as the age at which the groups appear to start diverging visually, but we also provide the lower age at which the confidence intervals (at 95%) between groups did not overlap.

The diagnostic performance of plasma p-tau181 and NfL was determined with receiver operating characteristic (ROC) curve analyses. DeLong's test was used to compare the different areas under the curve (AUC). To assess the correlation between plasma p-tau and other fluid biomarkers, Spearman correlation was used. P-tau181 values were non-normally distributed. Therefore, we present the raw data in the figures and ROC analyses although we used log-transformed the plasma p-tau181 values for the neuroimaging analyses.

All significance tests were two-sided with the statistical significance set at 5%. Finally, for the neuroimaging variables, we first performed vertex-wise correlation analyses between p-tau181 plasma and both CTh and FDG-PET uptake in the whole sample. We then conducted separate stratified analyses in asymptomatic and symptomatic AD participants. To correct for multiple comparisons, the threshold of significance was set at family-wise error (FWE)-corrected $P < 0.05$ for all vertex-wise analyses.

**Reporting summary**. Further information on research design is available in the Nature Research Reporting Summary linked to this article.

## Data availability

Anonymized data will be shared upon request from a qualified academic investigator for the sole purpose of replicating procedures and results of the article and as long as data transfer is in agreement with EU legislation on the general data protection and the transfer is approved by the Ethical Review Board of the Sant Pau Research Institute.

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

## Acknowledgements

This study was supported by the Fondo de Investigaciones Sanitario (FIS), Instituto de Salud Carlos III (PI14/01126 and PI17/01019 to J.F., PI13/01532 and PI16/01825 to R.B., PI18/00335 to M.C.I., PI18/00435 and INT19/00016 to D.A., PI15/01618 to R.R., PI14/1561 and AC19/00103 to A.L.) and the CIBERNED program (Program 1, Alzheimer Disease to A.L. and SIGNAL study, www.signalstudy.es), partly jointly funded by Fondo Europeo de Desarrollo Regional, Unión Europea, Una manera de hacer Europa. This work was also supported by the National Institutes of Health (NIA grants 1R01AG056850 - 01A1; R21AG056974 and R01AG061566 to J.F.), Departament de Salut de la Generalitat de Catalunya, Pla Estratègic de Recerca i Innovació en Salut (SLT002/16/00408 to A.L.), Fundació La Marató de TV3 (20141210 to J.F., 044412 to R.B. and 201437.10 to R.R.); I. Illán-Gala is supported by the Rio Hortega grant (CM17/00074) from "Acción Estratégica en Salud 2013-2016" and the Global Brain Health Institute (https://www.gbhi.org/). Fundació Catalana Síndrome de Down and Fundació Víctor Grífols i Lucas partially supported this work. This work was also supported by Generalitat de Catalunya (SLT006/17/00119 to J.F., SLT006/17/95 to E.V. and SLT006/17/00125 to D.A.) and a grant from the Fundació Bancaria La Caixa to R.B. M.F.I. acknowledges support from the Jérôme Lejeune and Sysley D'Ornano Foundations. A Bejanin was the recipient of a Juan de la Cierva-Incorporación grant from the Spanish Ministry of Economy and Competitiveness (IJCI-2017-32609) and a Miguel Servet I grant (CP20/00038) from the Carlos III Health Institute. T.K.K. holds a research fellowship from the BrightFocus Foundation (#A2020812F) and is further supported by the Swedish Alzheimer Foundation (Alzheimerfonden; #AF-930627), the Swedish Brain Foundation (Hjärnfonden; #FO2020-0240), the Swedish Dementia Foundation (Demensförbundet), Gamla Tjänarinnor Foundation, the Aina (Ann) Wallströms and Mary-Ann Sjöbloms Foundation, the Agneta Prytz-Folkes & Gösta Folkes Foundation (#2020-00124), the Gun and Bertil Stohnes Foundation, and the Anna Lisa and Brother Björnsson's Foundation. H.Z. is a Wallenberg Scholar supported by grants from the

Swedish Research Council (#2018-02532), the European Research Council (#681712), Swedish State Support for Clinical Research (#ALFGBG-720931), the Alzheimer Drug Discovery Foundation (ADDF), USA (#201809-2016862), the AD Strategic Fund and the Alzheimer's Association (#ADSF-21-831376-C, #ADSF-21-831381-C and #ADSF-21-831377-C), the Olav Thon Foundation, the Erling-Persson Family Foundation, Stiftelsen för Gamla Tjänarinnor, Hjärnfonden, Sweden (#FO2019-0228), the European Union's Horizon 2020 research and innovation programme under the Marie Skłodowska-Curie grant agreement No 860197 (MIRIADE), and the UK Dementia Research Institute at UCL. K.B. is supported by the Swedish Research Council (#2017-00915), the Alzheimer Drug Discovery Foundation (ADDF), USA (#RDAPB-201809-2016615), the Swedish Alzheimer Foundation (#AF-742881), Hjärnfonden, Sweden (#FO2017-0243), the Swedish state under the agreement between the Swedish government and the County Councils, the ALF-agreement (#ALFGBG-715986), and European Union Joint Program for Neurodegenerative Disorders (JPND2019-466-236). The authors would like to thank all the participants with Down's syndrome, their families and their carers for their support of, and dedication to this research. We also acknowledge the Fundació Catalana Síndrome de Down for global support; Soraya Torres, Shaimaa El Bounasri, Laia Muñoz and Raúl Núñez for laboratory and sample handling; Reyes Alcoverro, Marta Salinas and Tania Martínez for administrative support; Concepción Escola for nursing handling. We also thank the clinicians for their help in acquiring the data reported in this article.

## Author contributions

A.L. and J.F. did the literature search. J.F., H.Z., K.B. and A.L. conceived and designed the study. J.F., H.Z., T.K., M.C.I., N.A., V.M., I.B., L.V., M.A., B.B., S.F., S.V., D.A., K.B. and A.L. acquired data. J.F., H.Z., M.C.I., D.A., J.P., V.M., R.B., K.B. and A.L. analysed and interpreted the data. J.P. did the statistical analyses. J.F. and A.L. drafted the manuscript. All authors revised the manuscript for important intellectual content and edited the manuscript.

## Competing interests

Dr. Lleó has served as a consultant or at advisory boards for Fujirebio-Europe, Roche, Biogen and Nutricia. In addition, Dr. Lleó has a patent WO2019175379 A1 Markers of synaptopathy in neurodegenerative disease issued. Dr. Zetterberg has served as a consultant or at advisory boards for Denali, Roche Diagnostics, Wave, Samumed, Siemens Healthineers, Pinteon Therapeutics and CogRx, has given lectures in symposia sponsored by Fujirebio, Alzecure and Biogen, and is a co-founder of Brain Biomarker Solutions in Gothenburg AB (BBS), which is a part of the GU Ventures Incubator Program.
Dr. Alcolea has served as a consultant or at advisory boards for Krka Farmacéutica S.L., Fujirebio-Europe, Roche Diagnostics and Nutricia. In addition, Dr. Alcolea has a patent WO2019175379 A1 Markers of synaptopathy in neurodegenerative disease issued. Dr. Blennow has served as a consultant or at advisory boards for Abcam, Axon, Biogen, JOMDD/Shimadzu. Julius Clinical, Lilly, MagQu, Novartis, Roche Diagnostics, and Siemens Healthineers, and is a co-founder of Brain Biomarker Solutions in Gothenburg AB (BBS), which is a part of the GU Ventures Incubator Program. Dr. Fortea has served as a consultant or at advisory boards for AC Immune, Novartis, and Merck. In addition, Dr. Fortea has a patent WO2019175379 A1 Markers of synaptopathy in neurodegenerative disease issued. The other authors declare no competing interests.
