## [Peer Review File · Nature Communications]

Reviewers' Comments:

Reviewer #1:

Remarks to the Author:

The authors carefully describe their investigation of the value of plasma p-tau181 in the detection of AD and comparisons with established biochemical and imaging biomarkers in adult Down's syndrome patients. A few suggestions for the authors' consideration:

1. In Figure 2 C (as labelled in the manuscript) ROC curves are displayed for comparing the asymptomatic group with either the clinically diagnosed AD dementia group or the prodromal group. In sporadic AD studies it is well known that clinical diagnosis is limited and that up to about 20% of cases are not AD based upon various objective markers or autopsy. How accurate are these diagnoses in Down's syndrome patients? This potential caveat should be discussed.
2. In sporadic AD it is appreciated that unless the amyloid, tau and neurodegeneration status are taken into account in MCI patients, there can be a significant number of subjects who are amyloid negative but tau positive and neurodegeneration pos or neg. It is recommended to take this potential occurrence into account in assessing comparisons with amyloid PET pos and neg since some of the amyloid neg individuals may be CSF tau pos and cause some degree of discordance. It is recommended to include this consideration in reviewing the comparisons with amyloid PET.
3. "Standardized differences" are utilized in Fig 1B but what exactly this derived variable is is not clear to this reviewer and will help the reader to better understand what the authors are communicating here.
4. Minor comment: text and labelling differences in Figure 2 need to be corrected.

Reviewer #2:

Remarks to the Author:

Leo et al present their research investigating quantification of phosphorylated tau 181 (p-tau) in plasma in adults with Down syndrome (DS) as a biomarker of Alzheimer's disease (AD) pathology. Given the early stage of investigation of this biomarker in the DS population, this is a large study with 366 adults with DS (240 asymptomatic, 43 prodromal, 83 AD dementia) and 44 adult euploid controls (approximately mean age- and sex-matched, noting that the control group is older than the asymptomatic DS group). The performance of p-tau was compared to clinical diagnosis in the DS population and separately, for a subset of participants imaging metrics including, most critically Florbetapir PET (n=45). The gold standard comparator--autopsy evidence of AD pathology--was, understandably, not included in this study. The authors convincingly demonstrate the potential of plasma p-tau as a biomarker for AD pathology in DS, similar to previous findings demonstrating the performance of the biomarker in sporadic AD.

INTRODUCTION/DISCUSSION

It would be helpful for the authors to further describe the potential significance of a biomarker for AD pathology in a population with a lifetime risk > 90% for the disease. For example, the authors elude to underdiagnoses but provide no statistic for this.

METHODS

It is notable that in a paper about fluid biomarkers, the imaging protocol description is more detailed than the biofluid protocol description. A basic description of collection and processing procedure, range of storage times for each group, # of freeze/thaw cycles, is warranted in a manuscript about biomarkers.

DISCUSSION

Line 297: "...plasma p-tau 1818 levels... showed a two-fold increase in patients..." should be altered to clearly indicate that the authors are referring to the mean plasma p-tau concentration given the overlap between groups.

RELEVANT TO MULTIPLE SECTIONS OF THE MANUSCRIPT

It is recommended to change "level" to "concentration" in the text when discussing biomarker concentrations.

The authors commonly rely on subjective descriptive terms when discussing biomarker diagnostic accuracy. It would be preferable to state the numeric accuracy and let the readers decide if this is "accurate", "good diagnostic performance", "high accuracy" etc. For instance, I would not characterize an AUC of 0.80 as having "high accuracy for the diagnosis of AD in DS" (line 237), particularly as the authors are proposing this assay as a screening test for AD in this population.

It would be helpful to compare performance of plasma p-tau to the combination of other metrics such as age and apoE status. As is known from the non-DS population, these metrics can approach the diagnostic performance of some biofluid markers.

It would be helpful for the authors to address the overlap in plasma p-tau concentrations in the groups studied and the modest AUC, given their proposal to use this biomarker as a screening test.

The authors state in the abstract and elsewhere that the data "support the use of plasma p-tau 181 in clinical practice and clinical trials in this population." Based on the data presented p-tau 181 looks to be a compelling biomarker for further study in research applications (e.g., clinical trials). On the other hand, the authors have not demonstrated compelling data/discussions for use in clinical care. Greater articulation of this application and/or a more tempered discussion of the readiness for clinical use, would help better align the conclusions with the data presented.

FIGURES

Figure 2 is mislabeled.

Reviewer #3:

Remarks to the Author:

The manuscript by Lleo and colleagues adds to the growing knowledge and similarity of Alzheimer's disease biomarkers in people with Down syndrome (DS) compared to autosomal dominant and sporadic AD. This current study complements work led previously by Dr. Juan Fortea and published by many of the authors here, correlating AD biomarkers with clinical stages of AD in a very large and well-characterized cohort of DS individuals in Barcelona, Spain. Plasma p-tau181 levels were quantified by Simoa from samples collected from 366 adults with DS (240 asymptomatic, 43 prodromal, 83 AD dementia) and 44 euploid controls. Appropriate statistical analyses were employed. Phospho-tau181 was found to increase starting in the early 30's in DS individuals and was approximately 2-fold higher in DS participants with prodromal AD or AD dementia compared to those with asymptomatic AD. Plasma p-tau181 in participants with DS correlated with CSF p-tau181 and CSF total tau and, was inversely correlated with CSF Abeta42/40. Plasma p-tau181 also correlated with atrophy and hypometabolism in AD-relevant brain regions in DS individuals, especially in those who were in prodromal or AD dementia stages. The diagnostic accuracy of plasma p-tau181, which is AD-specific, was similar to the high diagnostic accuracy of plasma NfL, a non-AD-specific marker of neurodegeneration, in this population. These results suggest that plasma p-tau181 may provide good diagnostic accuracy and may be helpful in screening and monitoring of DS participants for clinical trials. This data bodes well for the utility of plasma p-tau181 for diagnosing and monitoring disease progression in all AD. Overall, this is a well-written, concise paper that builds upon prior data and supports the use of plasma biomarkers for AD. The strengths and limitations are clearly addressed. The need for future studies is underscored.

Comments:

1. Was there any relationship between intellectual disability level and plasma p-tau181 (or any other fluid or imaging biomarker)?
2. Why was genetic screening for DS only assessed in 308 of the 366 adults with DS?
3. APO E status is mentioned in the methods, but there is no mention of it in Table 1 or the results. Was there any relationship between APOE4 and plasma p-tau181, plasma NfL, amyloid PET, FDG-PET, MRI or any of the CSF biomarkers?
4. Abundant cerebellar Abeta deposits have been reported in older individuals with DS. Is it possible that using cerebellum as the reference region for amyloid PET imaging may confound the results?
5. Maybe I missed it but did plasma p-tau181 (which correlated with plasma NfL) correlate with CSF NfL in participants with DS, especially those with prodromal or AD dementia?

Minor Comments:

6. Typo? Third paragraph of Results: "Levels of plasma p-tau181 in participants with DA and AD dementia were higher compared to those of participants with prodromal AD". Should DA be DS?
7. AC Immune should be AC Immune.

Point-by-point response to the reviewers' comments

Reviewer #1

Comment 1. The authors carefully describe their investigation of the value of plasma p-tau181 in the detection of AD and comparisons with established biochemical and imaging biomarkers in adult Down's syndrome patients. A few suggestions for the authors' consideration:

1. In Figure 2 C (as labelled in the manuscript) ROC curves are displayed for comparing the asymptomatic group with either the clinically diagnosed AD dementia group or the prodromal group. In sporadic AD studies it is well known that clinical diagnosis is limited and that up to about 20% of cases are not AD based upon various objective markers or autopsy. How accurate are these diagnoses in Down's syndrome patients? This potential caveat should be discussed.

Response: We thank the reviewer for her/his positive comment. In our cohort the diagnosis of AD was made independently by the neuropsychologist and neurologist blinded to biomarker data. Conflicting cases were resolved by consensus as recommended by the National Task Group on Intellectual Disabilities and Dementia Practices Consensus Recommendations for the Evaluation and Management of Dementia in Adults with Intellectual Disabilities (<https://www.medpagetoday.com/upload/2013/8/20/PIIS0025619613003716.pdf>). We recognize that the clinical diagnosis of AD dementia, and especially prodromal AD, is challenging as the intellectual disability associated with DS complicates the diagnosis. However, DS is currently conceptualized as a genetic form of AD (Dubois et al, Lancet Neurol 2014). Therefore, in this population, AD is by far the main neurological comorbidity, being other causes exceedingly rare. The differential diagnosis, thus, does not commonly include other forms of neurodegenerative dementias. This fact explains the good performance of AD biomarkers in DS compared to the general population (Fortea et al, Lancet Neurol 2018). We would also like to note that the potential clinical misdiagnoses would reduce the diagnostic performance of AD biomarkers, which despite this potential confounder is very high. It is possible that when longer longitudinal follow-up is available in the participants, some diagnoses change, and the diagnostic performance of the biomarkers increase, especially in prodromal AD. This has been addressed in detail in previous papers of our group (Benejam et al, A&D (Amst) 2020). These aspects have been discussed in the revised manuscript (methods, & discussion, page 12, lines 12-15).

Comment 2. In sporadic AD it is appreciated that unless the amyloid, tau and neurodegeneration status are taken into account in MCI patients, there can be a significant number of subjects who are amyloid negative but tau positive and neurodegeneration pos or neg. It is recommended to take this potential occurrence into account in assessing comparisons with amyloid PET pos and neg since some of the amyloid neg individuals may be CSF tau pos and cause some degree of discordance. It

is recommended to include this consideration in reviewing the comparisons with amyloid PET.

Response: We thank the referee for raising this important comment. DS, similarly to autosomal dominant AD, is considered a genetic form of AD linked to A β overproduction due to triplication of the APP gene. With the exception of exceedingly rare cases of DS with deletions in the APP region (J Alzheimer's Dis. 2017;56:459-70), all DS cases have universal AD pathology in the fourth decade. In fact, in both DS and ADAD, abnormal biomarkers of A β pathology are detectable at very young ages in CSF and about 10 years later in amyloid PET (Fortea et al, Lancet 2020). Therefore, in contrast to sporadic AD, subjects without A β pathology but tau positive biomarkers are exceedingly rare in DS. In particular, in our cohort, of the 45 DS cases with amyloid PET, 10 were amyloid negative (SUVR<1.11). Of these 10, 9 have CSF available. Of these 9, 8 had a positive A β 42/40 ratio (2 of them had also increased p-tau levels just above our validated cut-off). We have added a comment in the results section of the revised manuscript (page 7, line 14) to incorporate these data.

Comment 3. "Standardized differences" are utilized in Fig 1B but what exactly this derived variable is not clear to this reviewer and will help the reader to better understand what the authors are communicating here.

Response: We apologize for this misunderstanding. We would like to clarify that in our manuscript we computed standardized differences by the difference between the DS and the control levels divided by the standard deviation of both groups. We have included a comment in the Fig. 1 legend to clarify this issue.

Comment 3. Minor comment: text and labelling differences in Figure 2 need to be corrected.

Response: we have corrected these differences in the revised manuscript.

Reviewer #2:

Leo et al present their research investigating quantification of phosphorylated tau 181 (p-tau) in plasma in adults with Down syndrome (DS) as a biomarker of Alzheimer's disease (AD) pathology. Given the early stage of investigation of this biomarker in the DS population, this is a large study with 366 adults with DS (240 asymptomatic, 43 prodromal, 83 AD dementia) and 44 adult euploid controls (approximately mean age- and sex-matched, noting that the control group is older than the asymptomatic DS group). The performance of p-tau was compared to clinical diagnosis in the DS population and separately, for a subset of participants imaging metrics including, most critically Florbetapir PET (n=45). The gold standard comparator--autopsy evidence of AD pathology--was, understandably, not included in this study. The authors

convincingly demonstrate the potential of plasma p-tau as a biomarker for AD pathology in DS, similar to previous findings demonstrating the performance of the biomarker in sporadic AD.

Response: we thank the reviewer for the positive remarks.

Comment 1.

INTRODUCTION/DISCUSSION

It would be helpful for the authors to further describe the potential significance of a biomarker for AD pathology in a population with a lifetime risk > 90% for the disease. For example, the authors elude to underdiagnoses but provide no statistic for this.

Response: We thank the reviewer for this comment. There is substantial clinical misdiagnosis and underdiagnosis of dementia (Taylor DH et al, J Clin Epidemiol. 2002;55:929–937; Taylor DH et al, J Alzheimer's Dis. 2009;17:807–815).

Underdiagnosis prevents timely access to treatment, and those (undiagnosed) patients receive fewer health services than those with diagnosed dementia. This is a special problem in the context of DS, as most AD cases are not diagnosed due to lack of awareness from families, caregivers and clinicians. This lack of awareness leads to fewer consultations for cognitive decline, which are often delayed until functional decline or behavioral problems occur. Furthermore, there are difficulties in the clinical diagnosis due to the underlying intellectual disability in this population. Minimizing underdiagnosis was the primary objective of our health plan, and one of the main strengths of our cohort, which is population-based and in which we offer free screening for cognitive decline (<https://www.t21rs.org/wp-content/uploads/2020/02/T21RS-Science-Society-Bulletin-2015-2.pdf>).

Misdiagnosis, was raised also by reviewer 1. In our cohort the diagnosis of AD was made independently by the neuropsychologist and neurologist blinded to biomarker data and conflicting cases were resolved by consensus as recommended by the National Task Group on Intellectual Disabilities and Dementia Practices Consensus Recommendations for the Evaluation and Management of Dementia in Adults with Intellectual Disabilities (<https://www.medpagetoday.com/upload/2013/8/20/PIIS0025619613003716.pdf>). We recognize that the clinical diagnosis of AD dementia, and especially prodromal AD, is challenging as the intellectual disability associated with DS complicates the diagnosis. However, DS is currently conceptualized as a genetic form of AD (Dubois et al, Lancet Neurol 2014). Therefore, in this population, AD is by far the main neurological comorbidity, being other causes exceedingly rare. The differential diagnosis does not include other forms of neurodegenerative dementias. This fact explains the good performance of AD biomarkers in DS compared to the general population (Fortea et al, Lancet Neurol 2018). We would like also to note that the potential clinical misdiagnoses would reduce the diagnostic performance of AD biomarkers, which despite this potential confounder is very high. It is possible that when longer longitudinal follow-up is available in the participants, some diagnosis change and the diagnostic performance

of the biomarkers increase, especially for prodromal AD. This has been addressed in detail in previous papers of our group (Benejam et al, A&D (Amst) 2020). These aspects have been discussed in the revised manuscript (introduction, methods, & discussion, page 4, lines 8-9; page 12, lines 12-15).

Comment 2. METHODS

It is notable that in a paper about fluid biomarkers, the imaging protocol description is more detailed than the biofluid protocol description. A basic description of collection and processing procedure, range of storage times for each group, # of freeze/thaw cycles, is warranted in a manuscript about biomarkers.

Response: We agree with the reviewer in that more information would be useful to readers. The inclusion of a brief statement in the original manuscript was due to the fact that our fluid collection and processing protocol has been previously published (Alcolea et al, 2019) and also cited in other publications about fluid biomarkers in DS (Fortea et al, Lancet 2018). We have now expanded this information in the revised version of the manuscript (methods section, page 13, lines 15-21). In particular, we have included data about the range of storage time and that all samples had a maximum of two freeze/thaw cycles before the analysis.

Comment 3. DISCUSSION

Line 297: "...plasma p-tau 181 levels... showed a two-fold increase in patients..." should be altered to clearly indicate that the authors are referring to the mean plasma p-tau concentration given the overlap between groups.

Response: We have rephrased this statement and have included "median plasma p-tau concentration" in the results section.

Comment 4. RELEVANT TO MULTIPLE SECTIONS OF THE MANUSCRIPT

It is recommended to change "level" to "concentration" in the text when discussing biomarker concentrations.

Response: We have changed "level" to "concentration" in the entire manuscript.

Comment 5. The authors commonly rely on subjective descriptive terms when discussing biomarker diagnostic accuracy. It would be preferable to state the numeric accuracy and let the readers decide if this is "accurate", "good diagnostic performance", "high accuracy" etc. For instance, I would not characterize an AUC of 0.80 as having "high accuracy for the diagnosis of AD in DS" (line 237), particularly as the authors are proposing this assay as a screening test for AD in this population.

Response: We agree with the reviewer's comment. We have now deleted all subjective terms when referring to diagnostic accuracy of p-tau along the revised manuscript.

Comment 6. It would be helpful to compare performance of plasma p-tau to the combination of other metrics such as age and apoE status. As is known from the non-DS population, these metrics can approach the diagnostic performance of some biofluid markers.

Response:

We thank the reviewer for the suggestion. We have now included these analyses in supplementary material (Suppl Fig. 2).

Specifically we performed ROC analysis to assess the diagnostic performance of p-tau (and NfL), age, *APOE* genotype and their combination to distinguish aDS from pDS, dDS and the combination of symptomatic patients (pDS+dDS).

ROC curves for plasma p-tau (top row) and NfL (bottom row) comparing the individuals with asymptomatic Down syndrome with prodromal Alzheimer’s disease (left), with demented Alzheimer’s disease (center) and symptomatic Alzheimer’s disease (right). aDS: asymptomatic Down syndrome; pDS: prodromal Alzheimer’s disease in Down syndrome; dDS: Alzheimer’s disease dementia in Down syndrome; AUC: area under the curve; CI: confidence interval; P-tau: baseline plasma p-tau levels; NfL: baseline plasma NfL levels

Comment 7. It would be helpful for the authors to address the overlap in plasma p-tau concentrations in the groups studied and the modest AUC, given their proposal to use this biomarker as a screening test.

Response: We agree with the reviewer in that there was significant overlap in the concentration of p-tau across groups. This is not entirely unexpected because we know that the degree of tau pathology varies among individuals with DS and similar degree of

dementia. However, the AUC of p-tau in our study are acceptable for a screening test given that we have other biomarkers (CSF or PET) that can be used to confirm the diagnosis. We have elaborated this issue in the results section of the manuscript (page 5, line 24).

Comment 8. The authors state in the abstract and elsewhere that the data “support the use of plasma p-tau 181 in clinical practice and clinical trials in this population.” Based on the data presented p-tau 181 looks to be a compelling biomarker for further study in research applications (e.g., clinical trials). On the other hand, the authors have not demonstrated compelling data/discussions for use in clinical care. Greater articulation of this application and/or a more tempered discussion of the readiness for clinical use, would help better align the conclusions with the data presented.

Response: we agree with this reviewer that at this point it is premature to state that p-tau181 can be used in clinical practice, given the number of assays and the long period needed to standardize a biomarker for clinical routine. We have rephrased the statements about implementation in clinical practice in the revised form of the manuscript.

Comment 9. FIGURES

Figure 2 is mislabeled.

Response: We apologize for this error that was also mentioned by reviewer 1. We have corrected this figure in the revised manuscript.

Reviewer #3 (Remarks to the Author):

The manuscript by Lleo and colleagues adds to the growing knowledge and similarity of Alzheimer’s disease biomarkers in people with Down syndrome (DS) compared to autosomal dominant and sporadic AD. This current study complements work led previously by Dr. Juan Fortea and published by many of the authors here, correlating AD biomarkers with clinical stages of AD in a very large and well-characterized cohort of DS individuals in Barcelona, Spain. Plasma p-tau181 levels were quantified by Simoa from samples collected from 366 adults with DS (240 asymptomatic, 43 prodromal, 83 AD dementia) and 44 euploid controls. Appropriate statistical analyses were employed. Phospho-tau181 was found to increase starting in the early 30’s in DS individuals and was approximately 2-fold higher in DS participants with prodromal AD or AD dementia compared to those with asymptomatic AD. Plasma p-tau181 in participants with DS correlated with CSF p-tau181 and CSF total tau and, was inversely correlated with CSF Aβ_{42/40}. Plasma p-tau181 also correlated with atrophy and hypometabolism in AD-relevant brain regions in DS individuals, especially in those

who were in prodromal or AD dementia stages. The diagnostic accuracy of plasma p-tau181, which is AD-specific, was similar to the high diagnostic accuracy of plasma NfL, a non-AD-specific marker of neurodegeneration, in this population. These results suggest that plasma p-tau181 may provide good diagnostic accuracy and may be helpful in screening and monitoring of DS participants for clinical trials. This data bodes well for the utility of plasma p-tau181 for diagnosing and monitoring disease progression in all AD. Overall, this is a well-written, concise paper that builds upon prior data and supports the use of plasma biomarkers for AD. The strengths and limitations are clearly addressed. The need for future studies is underscored.

Comments:

1. Was there any relationship between intellectual disability level and plasma p-tau181 (or any other fluid or imaging biomarker)?

Response:

Response: We thank the reviewer for the suggestion. There were significant differences among the different levels of intellectual disability (see figure below, Suppl Fig 1). This might be due to cognitive resilience in those with milder forms or to a later diagnosis in subjects with more severe intellectual disability, which might generate a Will Rogers phenomenon (ie. when moving an element from one set to another set raises the average values of both sets).

We have added this figure to the supplementary material, and we have included a sentence in the results section and discussion section (page 6, lines 1-3).

Baseline levels of plasma p-tau (a) and NfL plasma (b) across mild, moderate and severe/profound intellectual disability. * indicates p<0.05; ** indicates p<0.01; *** indicates p<0.001

2. Why was genetic screening for DS only assessed in 308 of the 366 adults with DS?

Response: Unfortunately, not all subjects included in the study had a karyotype available in our records. This could be explained by the age of some of the subjects. However, we do not consider this a limitation since the accuracy of clinical diagnosis of DS is extremely high (N Engl J Med. 2020;382(24):2344-52).

3. APOE status is mentioned in the methods, but there is no mention of it in Table 1 or the results. Was there any relationship between APOE4 and plasma p-tau181, plasma NfL, amyloid PET, FDG-PET, MRI or any of the CSF biomarkers?

Response: We thank the reviewer for this suggestion. We fully agree that it would be very interesting to analyze the effect of APOE genotype on plasma p-tau and other biomarkers. However, this dataset is the basis of another publication that is currently being submitted. This independent manuscript describes the effect of APOE on several fluid and imaging biomarkers, and contains a large set of data that exceeds the limits of the current manuscript and we would prefer not to include it. We have now included APOEε4 status in the Table 1 of the revised manuscript.

4. Abundant cerebellar Abeta deposits have been reported in older individuals with DS. Is it possible that using cerebellum as the reference region for amyloid PET imaging may confound the results?

Response: We thank the reviewer for raising this issue. We used the cerebellum as the reference region, as this structure is reported as one of the last brain regions to accumulate amyloid (Thal et al, Acta Neuropathol 2018; 136: 557-67) and is the most widely used in cross-sectional imaging studies. However, in longitudinal amyloid PET studies, in order to diminish the confounding effects of amyloid deposition in the cerebellum the eroded white matter is increasingly being used (J Nucl Med. 2015;56:567-74).

We have therefore reanalyzed the data using the white matter composite (see figure below). Using this reference region, we observed that the results did not change significantly for the discrimination of amyloid PET positivity. The AUC for the comparison was 0.81 (95% CI 0.68-0.95). To be consistent with most of the cross-sectional literature, we have, nonetheless, maintained in the revised manuscript the whole cerebellum as the reference region for the amyloid PET analyses.

Figure R1 (for re-review purposes only): Left: Plasma p-tau levels for Down syndrome subjects stratified by Amyloid- β PET status using WM composite as a reference region (cut-off=0.79). Right: ROC curve for plasma p-tau181 comparing individuals with Down syndrome with positive and negative amyloid- β PET.

5. Maybe I missed it but did plasma p-tau181 (which correlated with plasma NfL) correlate with CSF NfL in participants with DS, especially those with prodromal or AD dementia?

Response: We thank the reviewer for this comment. In the original version we showed the correlation in Suppl. Fig 1 (now Suppl. Fig 3) but we did not mention these data in the body of the manuscript. Plasma p-tau181 concentration correlated with CSF NfL concentration in patients with DS and symptomatic AD (prodromal and dementia $\rho=0.51$; $p<0.0001$) and in those with prodromal AD ($\rho = 0.59$; $p=0.0017$). There was no correlation in the group of aDS or dDS ($\rho = 0.15$, $p = 0.23$; $\rho = 0.29$, $p = 0.07$, respectively). We have added these data in the revised version of the manuscript (page 7, line 2).

Minor Comments:

6. Typo? Third paragraph of Results: “Levels of plasma p-tau181 in participants with DA and AD dementia were higher compared to those of participants with prodromal AD”. Should DA be DS?

7. AC Immune should be AC Immune.

Response: we apologize for these typos. We have corrected them in the revised version.

Reviewers' Comments:

Reviewer #1:

Remarks to the Author:

Authors have fully addressed questions raised after the review. The revised manuscript is acceptable for publication.

Reviewer #2:

Remarks to the Author:

The authors have done a thorough job of addressing the reviewers comments in this latest revision.

Reviewer #3:

Remarks to the Author:

The authors have responded thoroughly to all of the reviewers' comments and have added new data. In particular, this reviewer appreciates the re-analysis of the amyloid PET scans using white matter as the reference for comparison with cerebellum, which is known to accumulate amyloid in DS. The results showed no significant difference so the authors retained their original data and figure using cerebellum, as this is the more frequently used reference region for amyloid imaging.

In addition, the authors generated new data to look at correlations between intellectual disability level and ptau-181 levels as shown in Supplementary Figure 1.

It would be helpful to define sDS in the Supplementary Figure 2 legend. This likely refers to Down syndrome with symptomatic AD (prodromal and dementia).

The APOEε4 genotype has been added Table 1 and correlations added to Suppl Fig 2. It is noted that the authors are preparing a separate manuscript addressing the impact of APOE genotype in another manuscript.

Taken together, the authors' responses to the critiques combined with new data and toning down of language around the clinical utility (at this time) are impressive. This paper will be of interest to the field and important as an additional potential biomarker for AD in Down syndrome, the largest at-risk genetic predisposition for AD.

Point-by-point response to the reviewers' comments

REVIEWERS' COMMENTS

Reviewer #1 (Remarks to the Author):

Authors have fully addressed questions raised after the review. The revised manuscript is acceptable for publication.

Authors' response: We thank the reviewer comment.

Replacement reviewer 1 has suggested to include ROC curves for CSF biomarkers comparing clinical groups which could be included in the Supplementary Material. They also recommend CSF/plasma ptau comparison should be briefly commented on to help evaluate if CSF biomarker could still have clinical utility to confirm AD diagnostic after plasma ptau screening.

Authors' response: We have now included these data in Suppl. figure 3.

Reviewer #2 (Remarks to the Author):

The authors have done a thorough job of addressing the reviewers comments in this latest revision.

Authors' response: We thank the reviewer comment.

Reviewer #3 (Remarks to the Author):

The authors have responded thoroughly to all of the reviewers' comments and have added new data. In particular, this reviewer appreciates the re-analysis of the amyloid PET scans using white matter as the reference for comparison with cerebellum, which is known to accumulate amyloid in DS. The results showed no significant difference so the authors retained their original data and figure using cerebellum, as this is the more frequently used reference region for amyloid imaging.

In addition, the authors generated new data to look at correlations between intellectual disability level and ptau-181 levels as shown in Supplementary Figure 1.

It would be helpful to define sDS in the Supplementary Figure 2 legend. This likely refers to Down syndrome with symptomatic AD (prodromal and dementia).

The APOEε4 genotype has been added Table 1 and correlations added to Suppl Fig 2. It is noted that the authors are preparing a separate manuscript addressing the impact of

APOE genotype in another manuscript.

Taken together, the authors' responses to the critiques combined with new data and toning down of language around the clinical utility (at this time) are impressive. This paper will be of interest to the field and important as an additional potential biomarker for AD in Down syndrome, the largest at-risk genetic predisposition for AD.

Authors' response: We thank the reviewer comments. We have now defined sDS (symptomatic AD in DS) in the Suppl figure 2 legend in the revised version of the manuscript.